# VARMA-EGARCH Model for Air-Quality Analyses and Application in Southern Taiwan

**Edward Ming-Yang Wu [1] and Shu-Lung Kuo [2,*]** 

[1] College of Maritime, National Kaohsiung University of Science and Technology, Kaohsiung 811, Taiwan; edmywu@isu.edu.tw

[2] Department of Technology Management, The Open University of Kaohsiung, Kaohsiung 806, Taiwan

\* Correspondence: singsuey@ms28.hinet.net

**Abstract:** This study adopted the Exponential Generalized Autoregressive Conditional Heteroscedasticity (EGARCH) model to analyze seven air pollutants (or the seven variables in this study) from ten air quality monitoring stations in the Kaohsiung–Pingtung Air Pollutant Control Area located in southern Taiwan. Before the verification analysis of the EGARCH model is conducted, the air quality data collected at the ten air quality monitoring stations in the Kaohsiung–Pingtung area are classified into three major factors using the factor analyses in multiple statistical analyses. The factors with the most significance are then selected as the targets for conducting investigations; they are termed "photochemical pollution factors", or factors related to pollution caused by air pollutants, including particulate matter with particles below 10 microns ($PM_{10}$), ozone ($O_3$) and nitrogen dioxide ($NO_2$). Then, we applied the Vector Autoregressive Moving Average-EGARCH (VARMA-EGARCH) model under the condition where the standardized residual existed in order to study the relationships among three air pollutants and how their concentration changed in the time series. By simulating the optimal model, namely VARMA (1,1)-EGARCH (1,1), we found that when $O_3$ was the dependent variable, the concentration of $O_3$ was not affected by the concentration of $PM_{10}$ and $NO_2$ in the same term. In terms of the impact response analysis on the predictive power of the three air pollutants in the time series, we found that the asymmetry effect of $NO_2$ was the most significant, meaning that $NO_2$ influenced the GARCH effect the least when the change of seasons caused the $NO_2$ concentration to fluctuate; it also suggested that the concentration of $NO_2$ produced in this area and the degree of change are lower than those of the other two air pollutants. This research is the first of its kind in the world to adopt a VARMA-EGARCH model to explore the interplay among various air pollutants and reactions triggered by it over time. The results of this study can be referenced by authorities for planning air quality total quantity control, applying and examining various air quality models, simulating the allowable increase in air quality limits, and evaluating the benefit of air quality improvement.

**Keywords:** air pollutant control area; air pollutants; photo chemical pollution factor; impact response analyses; multiple statistical analyses

## 1. Introduction

According to air quality monitoring reports from the Environmental Protection Administration, major pollutants causing the Pollutant Standard Index (PSI) [1] to exceed the air quality standard are particulate matter ($PM_{10}$) and ozone ($O_3$). Taiwan's air quality data are released to the public in the form of pollution standards index (PSI) values, following the procedure identical to that of the United States Environmental Protection Agency. The nation's PSI was formulated and promulgated in 1994 by the Environmental Protection Administration (EPA) based on the daily concentrations

of five air pollutants: particulate matter with particles below 10 microns ($PM_{10}$), sulfur dioxide ($SO_2$), nitrogen dioxide ($NO_2$), carbon monoxide ($CO$), and ozone ($O_3$). Air quality in central and southern Taiwan has deteriorated, especially after large-scale pollution sources were set up one after another. As population density continues to rise in tandem with the development of industries and transportation, it results in many pollutants affecting humans' habitats; consequently, people have become more aware of pollution treatments [2,3]. Among the treatments, air quality management stands out as an imperative issue. Since air is highly dispersive or can be transported long range, if the total emission tolerance of pollutants in one area is not kept below that area's capacity, the pollutants will not only influence the air quality in that area, but also in adjacent regions [4,5].

Air pollution is a well-known environmental problem associated with urban areas around the world [6,7]. Various monitoring programs have been used to determine air quality by generating vast amounts of data on the concentration of each of the previously mentioned air pollutant in different parts of the world. The large data sets often do not convey air quality status to the scientific community, government officials, policy makers, and in particular to the general public in a simple and straightforward manner [5]. Up to now, PSI has been developed and disseminated by many agencies in the U.S. Canada, Europe, Australia, China, Indonesia, Taiwan, etc.

Models analyzing the degree of fluctuation can largely be divided into two categories: predictable volatility and unpredictable volatility (or stochastic volatility). The former belongs to the Generalized Autoregressive Conditional Heteroscedasticity (GARCH) group (which includes the exponential GARCH (EGARCH) model adopted in our study), and can predict the fluctuations in the current term using all of the information acquired from the previous term. The biggest difference between the two models lies in their estimation procedures [8,9]. The EGARCH model is similar to the GARCH model as it can also capture other properties of a financial time series, such as volatility clustering. If the fluctuation rate at *t*-1 is high, it will be high at *t* as well. In other words, an impulse occurring at *t*-1 also has an effect on the fluctuation rate at *t*. Moreover, since the value adopted in the formula is the logarithm of the variance rather than the variance per se, there is no need to impose any limits on the parameters in the EGARCH model. As a result, the EGARCH model automatically meets the requirement that the variance must be positive, which is one of its main advantages. In general, applying maximum likelihood estimation under conditions allowed by the model helps to increase the optimization speed and makes the optimization results more reliable [8,10].

This study analyzed the data obtained from the ten automatic air quality monitoring stations in the Kaohsiung–Pingtung Air Pollutant Control Area via factor analysis for multivariate statistics, in order to determine the air pollution factors that most influenced the air quality in the Kaohsiung–Pingtung area, namely the photochemical pollution factors ($PM_{10}$, $O_3$, and $NO_2$). Next, on the condition that a standardized residual existed, this study explored how air quality in the Vector Autoregressive Moving Average (VARMA)-EGARCH model fluctuated according to seasons. Since the analysis of results can reflect the relationships among air pollutants in real time, the authorities can refer to the results when applying and examining various air quality models, simulating the allowable increase in air quality limits, and evaluating the benefits of air quality improvement.

## 2. Experimental Method and Methodology

### 2.1. Selection of Air Quality Monitoring Stations

The Kaohsiung–Pingtung Air Pollutant Control Area, established by the Environmental Protection Administration in southern Taiwan, consists of ten ordinary air quality monitoring stations (Meinong station, Nanzi station, Qianjin station, Renwu station, Zuoying station, Xiaogang station, Daliao station, Linyuan station in Kaohsiung, and Pingtung station and Chaozhou station in Pingtung). In these ten ordinary air quality monitoring stations, Meinong, Pingtung, Daliao, and Chaochou are located in sub-urban belonging to non-industrial zone; Nanzi, Renwu, Zuoying, Xiaogang and Linyuan belong to industrial zone; Cianjin is located in the urban area.

The ten air quality monitoring stations are located on the Pingtung Plain, scattered across areas from 30 to 100 m above sea level. The mean annual temperature of the area is from 24 to 25 °C; the average temperature of the warmest months (July to August) exceeds 30 °C, while that of the coldest months (January to February) remains above 18 °C. The average annual rainfall is between 1500 and 2000 mm. We chose the Kaohsiung–Pingtung Air Pollutant Control Area because this region includes Kaohsiung City, which suffers from the most severe air pollution in Taiwan. The city, home to approximately 2.77 million people, has eight industrial areas and a large steel plant. We chose this Air Pollutant Control Area as our research subject because the air quality is usually poor due to its industrial activities in addition to the large number of cars and motorcycles in the city.

We collected 420 complete air pollutant monitoring statistics between 1 January 2019 and 30 May 2020 and analyzed seven pollutants (or seven variables): $SO_2$, $NO_2$, CO, $PM_{10}$, $O_3$, THC, and $CH_4$ (please refer to Figure 1 for the locations of the ten air quality monitoring stations chosen for this study). These statistics were acquired by analyzing data collected over 24 h by the EPA's autonomous monitoring stations in the area. The objective in studying the statistics of up to one and a half years is to reflect the temporality effect on air quality; thus, we were able to observe the EGARCH model during different periods with different observation frequencies and elucidate whether variables in different time series present different information. The tool adopted for corroboration was EVIEWS 10.0

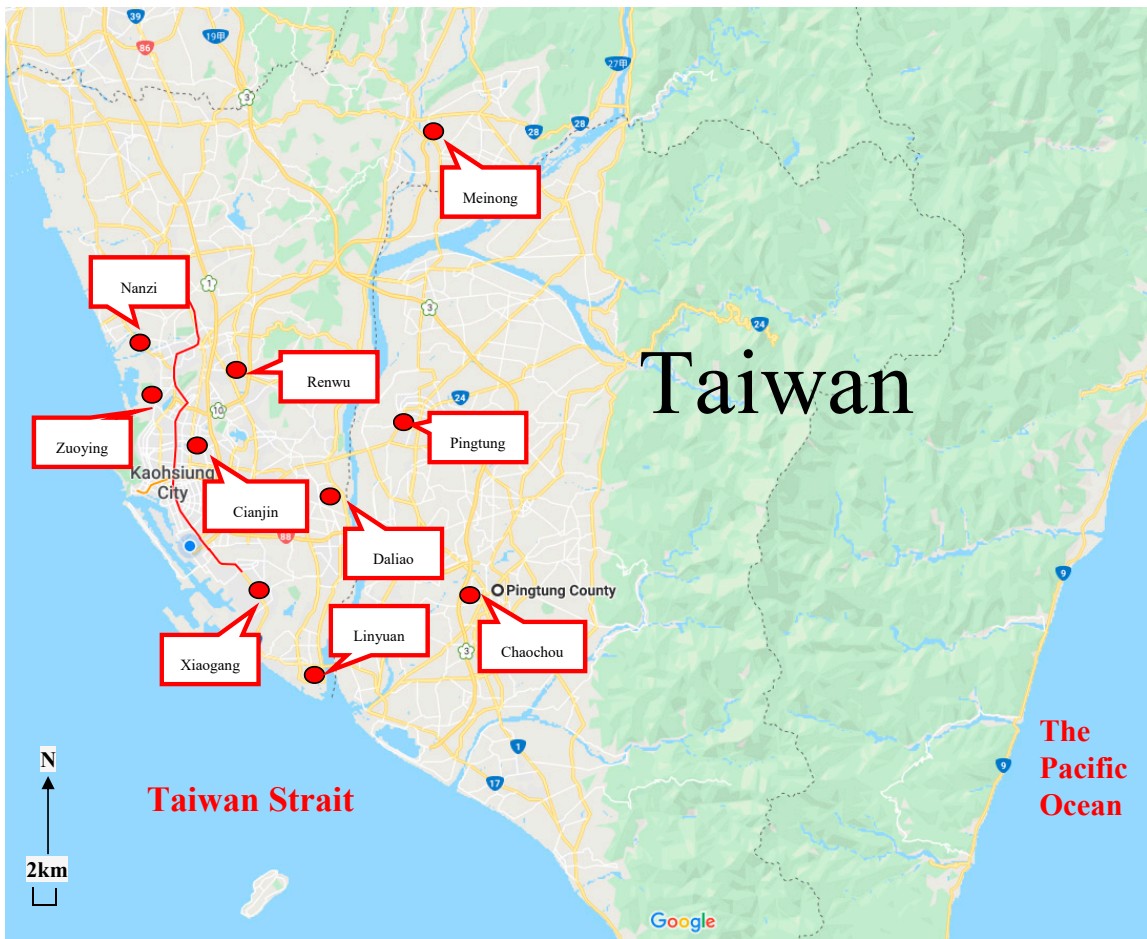

**Figure 1.** Air quality monitoring locations in Kaohsiung–Pingtung area, Taiwan.

## 2.2. Data Selection and Compilation

Prior to the analysis of the EGARCH model, we analyzed three factors using factor analyses in multiple statistical analyses that influences air quality in the Kaohsiung–Pingtung area: the photochemical pollution factor, the fuel pollution factor, and the organic pollution factor, employing

factor analysis for multivariate statistics. Next, we selected the photochemical pollution factor, which is the most influential factor determining air quality in the Kaohsiung–Pingtung area, to study the VARMA-EGARCH model.

*2.3. ARIMA Modeling*

A time series $\{x_t; \; t = 0, \pm 1, \pm 2, \; \ldots \}$ is ARMA $(p, q)$ if it is covariance stationary and can be represented as

$$x_t = \varphi_1 x_{t-1} + \ldots + \varphi x p_{t-p} + \varepsilon_t + \theta_1 \varepsilon_{t-1} + \ldots + \theta_q \varepsilon t - q \tag{1}$$

where $\varphi_p \neq 0$, $\theta_q \neq 0$, and $\varepsilon_t$ are the innovations with $N(0, \sigma_\varepsilon^2)$ and $\sigma_\varepsilon^2 > 0$. The parameters $p$ and $q$ are called the autoregressive [AR($p$)] and the moving average [MA($q$)] orders, respectively. When a time series does not appear covariance stationary, the differencing procedure may be applied to make it stationary. Then, the ARMA $(p, q)$ model can be applied to the stationary differenced time series and model so constructed is called ARIMA $(p, d, q)$ model where $d$ denotes the order of differencing [11,12]. The parameters $\varphi$ and $\theta$ have been estimated using maximum likelihood method in the present study.

An inspection of autocorrelation function (ACF) and partial autocorrelation function (PACF) helps in identifying the orders AR($p$) and MA($q$). In addition, more objectively defined criterions such as Akaike Information Criterion (AIC), Hannan–Quinn Information Criterion (HQIC), Bayesian Information Criterion (BIC) and Final Prediction Error (FPE) can also be used to identify the correct orders $p$ and $q$ [12,13].

*2.4. ARMA-EGARCH Modeling*

Conditional mean formula:

$$r_t = a_0 + \sum_{i=1}^{m} a_i r_{t-i} + \sum_{j=1}^{n} b_j E_{t-j} \; , \; E|\Omega_{t-1} \sim N(0, h_t)$$

Conditional variance formula:

$$\ln(h_t) \; = \; \alpha_0 + \sum_{i=1}^{q} \left( \gamma_i \frac{E_{t-i}}{\sqrt{h_{t-i}}} + \alpha_i \left( \frac{|E_{t-i}|}{\sqrt{h_{t-i}}} - E\left( \frac{|E_{t-i}|}{\sqrt{h_{t-i}}} \right) \right) \right) + \sum_{j=1}^{p} \beta_j \ln(h_{t-j}) \tag{2}$$

$\frac{E_{t-i}}{\sqrt{h_{t-i}}} > 0$ represents good news on the market while $\frac{E_{t-i}}{\sqrt{h_{t-i}}} < 0$ reflects bad news on the market; where, $\gamma_i = 0$ indicates that the degree of air pollution has a symmetrical effect in reaction to news impact.

If $\gamma_i < 0$, it suggests that the degree and fluctuation of air pollution incurred by air pollution shock are more significant than the situation when air pollution shock does not result in air pollution; in other words, a leverage effect can be observed when $\gamma_i < 0$. To determine whether the three GARCH models are applicable to the time series analysis, we first needed to confirm whether the time series has an ARCH effect; this was tested in this study using the Lagrange Multiplier (LM) test proposed by Engle [14]. Since it is necessary for the residual of the conditional mean formula in the GARCH model to follow the white noise process, all possible orders $(p, q)$ of the GARCH model must be compared with each other via trial and error. To determine whether or not residuals reached white noise, we later adopted the modified Q-statistics proposed by Ljung and Box [15] to analyze the residual of each series. However, the p and q orders of the model selected via trial and error might be subject to over-fitting. To solve this issue, we used the Akaike Information Criteria (AIC) [16] and Schwartz Bayesian Criterion (SBC) [17] to analyze the model under the principle of parsimonious parameterization.

### 2.5. Setting of the Model

This study used the following statistical principles and methods to conduct its simulation and predictive modeling for the selected photochemical pollution factors. The purpose was to explain the essence and meanings of the fat tail test, the Ljung–Box sequence test, and the ARCH test.

#### 2.5.1. Fat Tail Test

The main argument in this section is related to the last argument of the previous section—in the light of a non-explained empirical fact, it is preferable to try to describe it first with the available tools before increasing the model complexity. There are two main approaches to explain fat tails through Gaussian-based models. First, the fat tail test appears because the volatility of asset returns is dynamic. Therefore, looking at the unconditional distribution as described by histogram plots or kernel density plots, we can see heavy tails but they are mostly due to the time varying volatility. Second, fat tails appear because asset returns depend, in a non-linear fashion, on other factors which are distributed according to a Gaussian law. Results of examining the skewness, kurtosis, and Jarque–Bera normal distribution can be used for determining whether the distribution of modeling errors has fat tails.

#### 2.5.2. Ljung–Box Sequence Test

It was necessary to test whether the residual items in the regression model have sequence correlation before estimating the ARCH and (E)GARCH models. If the residual items have sequence correlation, the squared residual items will be examined to see if it has an ARCH effect. As such, it is very important to check if the residual items have sequence correlation before estimating the ARCH and (E)GARCH models.

#### 2.5.3. Examination of the ARCH Effectiveness

If the standard deviation of a time series is stationary, we can say that the variance of the time series is homogeneous, or heterogeneous, or vice versa. Before applying data to a heterogeneous variance model, we should examine data to see if they contain heterogeneous variance. This study adopted the Lagrange Multiplier (LM) proposed by Engle (1982) and Ljung and Box Q statistics [14] to test whether the variance of the time series is heterogeneous.

#### 2.5.4. Impact Response Analyses

The impulse response function (IRF) is used to study the impact of the structural disturbances on the variables in a VAR model over time; in other words, other variables' dynamic response pattern to an exogenous impulse when the said impulse impacts a variable. This study adopted the Cholesky method in dealing with orthogonalization, in order to analyze the intertemporal dynamic effects of indices across nations in the model, and the degree of dynamic interplay among those indices. The formulae are as follows:

$$Y_t = u + \sum_{i=0}^{\infty} \Phi_i \varepsilon_{t-i} \Phi_i = \mathrm{I}, \ i = 1, 2, \ldots, \infty \tag{3}$$

In Formula (3), $i$ represents the variable responding to an impulse. Error $\varepsilon_{t-1}$ can be interpreted as unexpected impulses in $t - i$. To examine how an impulse of a one-term delay influences the current term variables, one can refer to Formula (4).

$$\frac{@Y_t}{@E_{t-1}} = \Phi_{j,k,z} \tag{4}$$

In the above formula, the coefficient factor on row $j$, line $k$, formed a consecutive function over time, called the "impulse response function".

This study adopted the AIC proposed by Akaike [16] to select suitable lagged variables, which were represented as follows:

$$AIC(m) = T \times \ln(SSR/T) + 2m \qquad (5)$$

where, m indicates the number of variables in the model, T represents the number of samples of the three air pollutants, and SSR shows the sum of squares of the error terms.

## 3. Results and Discussion

### 3.1. Application of the Results of Factor Analysis

As mentioned above, this study divided air quality in the Kaohsiung–Pingtung area into three major factors according to the results of the factor analysis for multivariate statistics. It then chose the most important factor, the photochemical pollution factor, as its subject. This factor included three air pollutants: $PM_{10}$, $O_3$, and $NO_2$ (listed by the degree of their factor loadings). Moreover, since the data series had seasonality and cyclicity, the seven pollutant variables were standardized during factor analysis. The formula for standardization is as follows:

$$Z_{v,t} = \frac{Y_{v,t} - \mu_t}{\sigma_t} \; t \; = \; 1, \; \ldots \ldots \; w$$

where, $Y_{v,t}$ represents the original series, and $\mu_t$, $\sigma_t$ indicate the average and standard deviation in period 1~w.

### 3.2. Simulations of the Photochemical Pollution Factor with Models

3.2.1. Analysis of the Basic Properties of the Three Air Pollutants

The three air pollutants selected for the photochemical pollution factor of this study were $PM_{10}$, $O_3$, and $NO_2$. Table 1 presents the analysis results of the three air pollutants' basic properties between 1 January 2019 and 30 May 2020, including: average, standard deviation, skewness, kurtosis, and statistics obtained from the Jarque–Bera normality test. Firstly, in terms of kurtosis, the kurtosis coefficients of the three air pollutants were larger than the coefficient of normal distribution (which was 2), suggesting that each variant had the characteristic of a seasonal time series (in other words, pollutant concentrations vary according to seasons). In terms of skewness, the three pollutants' skewness leaned to the right (meaning that their skewness coefficients were all positive). Among them, $NO_2$ had the highest value of 4.3 when compared with $PM_{10}$ and $O_3$, which had values of 2.3 and 0.8, respectively (the smaller the value, the more stable the concentration). This result indicates that the concentration of $NO_2$ normally remains low. However, many statistics showed sudden spikes in the past, which corresponded to the results of Kuo and Ho [18]. This corroborated with the view that $NO_2$ had the strongest asymmetric effect, meaning that the concentration and degree of fluctuation of $NO_2$ formed in Kaohsiung were lower than those of the other two pollutants. Since $NO_2$ is least affected by the change of seasons, it is more difficult to predict its concentration fluctuations. Air pollutants that are non-compliant with the air quality standard in Kaohsiung and Pingtung are mainly $PM_{10}$ and $O_3$; hence, when the concentrations of $PM_{10}$ and $O_3$ increase, air pollution becomes more severe, especially during the winter. This result is reflected by the skewness of the two in Table 1, which is not high. Regarding $NO_2$, an increase in its concentration can only be observed in Kaohsiung and Pingtung during certain periods in winter and early spring; since the usual contribution of $NO_2$ to air pollution was less than $PM_{10}$ and $O_3$, it had a higher skewness coefficient. Kaohsiung City, home to approximately 2.77 million people, has eight industrial areas and a large steel plant. Air quality is usually poor due to the industrial activities in those areas and the large number of cars and motorcycles in the region. In addition, the Kaohsiung–Pingtung area is susceptible to pollutants from external sources brought by the north-eastern seasonal wind, and pollutants traveling from northern and central Taiwan. To make matters worse, pollutants are trapped in the area by an inversion caused by the

seasonal wind as it descends from the Central Mountain Range. In addition, emissions from factories and cars concentrated in Greater Kaohsiung make poor air quality more likely in the region during winter. Base on the above reasons, the air pollution in this area has been quite serious over the years.

**Table 1.** Basic properties of the three pollutants in the photochemical pollution factor.

| Items | $O_3$ (ppb) | $PM_{10}$ (μg/m³) | $NO_2$ (ppb) |
|---|---|---|---|
| Mean | 35.7 | 83 | 63.1 |
| Median | 32.6 | 78 | 59.3 |
| Maximum | 204 | 427 | 488 |
| Minimum | 8.6 | 20 | 16.6 |
| Std. Dev. | 0.8 | 0.8 | 0.4 |
| Kurtosis | 7.5 | 11.7 | 6.1 |
| Skewness | 0.8 | 2.3 | 4.3 |
| Jarque–Bera | 2518 | 2054 | 796 |
| Probability | 0.0 | 0.0 | 0.0 |
| Sum Sq. Dev. | 112.6 | 124.4 | 88.3 |
| Observations | 420 | 420 | 420 |

Table 1 also shows that the statistics obtained from the Jarque–Bera normality test are higher than the critical value (degree of freedom is 2, $X^2_{0.05,2}$ = 5.99) at a 5% significance level, reflecting the hypothesis that all variants refuse normal distribution. This means that all the three air pollutants had two fat tails, which proves that seasonality had a significant effect on them.

3.2.2. Examination of ARCH Effectiveness

The LM (Lagrance Multiplier) statistics [19] can be applied to examine whether the ARCH effect exists in a number sequence. The LM statistics is $TR^2$ with T being the number of samples, and $R^2$ being the determination coefficient value obtained using the ordinary least squares (OLS) regression; $T \times R^2$ obeys the chi-square distribution with P degrees of freedom. When the model LM statistics is obvious, the ARCH effect exists in the number sequence. The statistics of three air pollutants listed in Table 2 indicates that the conditional variance of the three air pollutants shows a strong ARCH effect (that is, all $T \times R^2$ values were significant at a 5% significant level). Hence, the ARCH effectiveness is appropriate for explaining these three air pollutants.

**Table 2.** Results of Autoregressive Conditional Heteroscedasticity (ARCH) effectiveness for 3 air pollutants.

| Q | $O_3$ (ppb) | $PM_{10}$ (μg/m³) | $NO_2$ (ppb) | Critical Value |
|---|---|---|---|---|
| (Lagged Variables) | (TR²) | (TR²) | (TR²) | $x^2_{(0.05,k)}$ |
| 1 | 15.57 | 356.21 | 140.36 | 3.84 |
| 2 | 18.98 | 367.29 | 154.22 | 5.99 |
| 3 | 21.22 | 381.00 | 163.87 | 7.82 |
| 4 | 24.58 | 390.54 | 179.78 | 9.49 |
| 5 | 30.16 | 393.58 | 192.65 | 11.07 |
| 6 | 33.20 | 402.63 | 216.60 | 12.59 |
| 7 | 36.61 | 411.29 | 231.47 | 14.07 |
| 8 | 38.14 | 416.85 | 249.54 | 15.51 |
| 9 | 39.52 | 438.20 | 266.33 | 16.92 |
| 10 | 42.65 | 468.13 | 289.51 | 19.68 |

Note: All TR² values less than 5% indicate "obviousness".

### 3.2.3. Ljung–Box Sequential Examination

Whether the residual in the regression model has serial correlation must be examined before the ARCH and EGARCH models are estimated. If the residual has serial correlation, the square of residual appears to have the ARCH and EGARCH effects. In this study, such examination was carried out using the Ljung–Box examination method; the results are listed in Table 3. All the examination statistics for L-B-Q (K) are smaller that the critical value so that null hypothesis, which is not conforming to alternative hypothesis, cannot be rejected. Hence, residuals of the various number sequences do not have serial correlation; this confirms to the phenomenon of white noise so that the model disposition is appropriate for these 3 air pollutants.

**Table 3.** Ljung–Box Sequence Test for 3 air pollutants.

| L-BQ (K) | $O_3$ (ppb) | $PM_{10}$ ($\mu g/m^3$) | $NO_2$ (ppb) | Critical Value $x^2_{(0.05,k)}$ |
|---|---|---|---|---|
| 1 | 1.92 | 0.84 | 0.57 | 3.84 |
| 2 | 2.62 | 2.01 | 0.95 | 5.99 |
| 3 | 5.18 | 3.36 | 2.64 | 7.82 |
| 4 | 6.79 | 5.69 | 4.26 | 9.49 |
| 5 | 9.00 | 7.17 | 6.61 | 11.07 |
| 6 | 10.48 | 8.60 | 8.47 | 12.59 |
| 7 | 12.46 | 11.43 | 10.69 | 14.07 |
| 8 | 12.89 | 12.52 | 11.12 | 15.51 |
| 9 | 13.62 | 13.77 | 12.06 | 16.92 |
| 10 | 16.31 | 15.95 | 12.95 | 18.31 |

To run the Ljung Box test by hand, we must calculate the statistic $Q$. For a time series $\gamma$ of length $n$:

$$Q(m) = n(n+2) \sum_{j=1}^{m} \frac{\gamma_j^2}{n - j'} \tag{6}$$

where: $\gamma_j$ = the accumulated sample autocorrelations, $m$ = the time lag.

We reject the null hypothesis and say that the model shows lack of fit if

$$Q > X^2_{1-\alpha,h}$$

where $X^2_{1-\alpha,h}$ is the $1 - \alpha$ quantile of the chi-square distribution with $h$ degrees of freedom.

### 3.2.4. Choosing the Best EGARCH Model

In order to subsequently simulate the best VARMA-EGARCH model, we tested different combinations with vector models VARMA and EGARCH. We chose the best one from the multiple VARMA (*p*, *q*)-EGARCH (*p*, *q*) models we created for the simulated analysis. Table 4 presents the analysis results. Among the models, VARMA (1,1)-EGARCH (1,1), the one with the smallest AIC and SC values, was chosen as the best model because it could best capture the fluctuations of air quality in different seasons.

**Table 4.** Analysis results of the best combination of photochemical pollution factor Vector Autoregressive Moving Average (VARMA) model and Exponential Generalized Autoregressive Conditional Heteroscedasticity (EGARCH) model.

| EGARCH Type<br>Vector Model | EGARCH (0,1) | | EGARCH (0,2) | | EGARCH (1,1) | | EGARCH (2,1) | |
|---|---|---|---|---|---|---|---|---|
| | AIC | SC | AIC | SC | AIC | SC | AIC | SC |
| VARMA (1,0) | 8.121 | 8.136 | 8.053 | 8.062 | 7.713 | 8.020 | 7.952 | 7.993 |
| VARMA (2,0) | 8.054 | 8.101 | 7.959 | 8.016 | 7.620 | 7.795 | 8.051 | 8.071 |
| VARMA (0,1) | 8.012 | 8.077 | 7.994 | 8.023 | 7.602 | 7.793 | 7.952 | 8.032 |
| VARMA (0,2) | 7.953 | 7.998 | 7.921 | 7.965 | 7.577 | 7.708 | 7.893 | 8.011 |
| VARMA (1,1) | 7.902 | 7.926 | 7.865 | 7.915 | 7.523 | 7.542 | 7.621 | 7.779 |
| VARMA (2,1) | 7.883 | 7.903 | 7.839 | 7.868 | 7.531 | 7.603 | 7.659 | 7.724 |

*3.3. Simulation Results of the Photochemical Pollution Factor VARMA (1,1)-EGARCH (1,1) Model*

According to the results of an extra factor analysis for multivariate statistics, factors under the photochemical pollution factor can be listed, with their factor loadings from big to small, as $P_{10} > O_3 > NO_2$. Photochemical reactions in the atmosphere are mainly triggered by radiation given off by the sun. Once pollutants (or precursors) absorb photons and enter an electronically excited state, they react with other pollutants and form $O_3$. In light of this phenomenon, this study designated $O_3$ as the dependent variable and $PM_{10}$ and $NO_2$ as independent variables and applied them to the VARMA (1,1)-EGARCH (1,1) model, in order to study how they change in the time series. Table 5 presents the relativity and trend of concentration of $PM_{10}$, $O_3$, and $NO_2$ in the time series under simulation with the VARMA (1,1)-EGARCH (1,1) model. This vector model can prevent bias when estimating conditional variance because it reflects the structural changes of the three air pollutants in different seasons.

The simulation results in Table 5 show that the concentration of $O_3$ (the t-statistic of $b_0$ was −0.75, meaning that it did not reach the significance level since it was <1.96) in the term could not be estimated according to the concentration of $PM_{10}$ in the same term. In contrast, the concentration of $PM_{10}$ with a lag time of one and two terms influenced the concentration of $O_3$ in the current term (the t-statistics of $b_1$ and $b_2$ were 2.56 and 2.04, respectively, meaning that they reached the significance level since they were >1.96). In terms of $NO_2$, we found that the concentration of $NO_2$ in the current term could not be used to estimate the concentration of $O_3$ in the same term (the t-statistic of $c_0$ was 0.32, meaning that it did not reach the significance level since it was <1.96). However, the concentration of $NO_2$ with a lag time of one term influenced the concentration of $O_3$ in the current term (the t-statistic of $c_1$ was 3.79, meaning that it reached the significance level since it was >1.96). Moreover, the concentration of $NO_2$ with a lag time of two terms still influenced the concentration of $O_3$ in the current term (the t-statistic of $C_2$ was 2.92, meaning that it reached the significance level since it was >1.96). However, the effect of $NO_2$ with a lag time of two terms on the concentration of $O_3$ was not as remarkable as that of $NO_2$ with a lag time of one term. From the above analyses of the three air pollutants, we found that the concentration of $O_3$ was not affected by that of $PM_{10}$ and $NO_2$ in the same term, but after a lag time of one term, the concentration of $O_3$ was influenced by $PM_{10}$ and $NO_2$ with a lag time of one to two terms. The results help to explain why $PM_{10}$ and $NO_2$ are the most important precursors for photochemical reactions in the atmosphere. Upon being released into the air from various sources of emission, $PM_{10}$ and $NO_2$ do not immediately start photochemical reactions with the sun; rather, they only start reacting with the sun after a one-term lag and form the final product, $O_3$. The photochemical reactions continue into the second term because $PM_{10}$ and $NO_2$ last longer in the atmosphere, thus being able to produce $O_3$ in the second term. However, from the simulation results in Table 5, it is found that the t-statistics of $b_2$ and $c_2$ (2.04 and 2.92, respectively) with a lag time of two terms reached the significance level, but their significance was less remarkable than that of $b_1$ and $c_1$ (2.56 and 3.79, respectively) with a lag time of one term. From the results, we can conclude that when $PM_{10}$ and $NO_2$ form in the atmosphere, we cannot use their concentrations to estimate

the concentration of $O_3$ in the current term. Rather, we must wait until they have been involved in photochemical reactions for a while, as that is when $O_3$ starts to form. In other words, only $PM_{10}$ and $NO_2$ with one- or two-term lag can influence the formation of $O_3$ in the current term. Furthermore, one-term lagged $PM_{10}$ and $NO_2$ have stronger impacts on the concentration of $O_3$ than those with a lag time of two terms. Regarding $O_3$, its concentration in the current term was significantly impacted by $O_3$ with a lag of one term (the t-statistic of $a_1$ was 7.03, meaning that it reached the significance level since it was >1.96), but was not significantly impacted by $O_3$ with a lag of two terms (the t-statistic of $a_2$ was 1.14, meaning that it did not reach the significance level since it was <1.96). These results suggested that although the concentration of $O_3$ in the current term is affected by that of $O_3$ with a one-term lag, the concentration in the current term is less impacted by $O_3$ with a two-term lag, as photochemical reactions start to decrease when the day proceeds into the evening. In this study, Kaohsiung–Pingtung Air Pollutant Control Area is located in the tropics. The annual average temperature is greater than 20 °C, and the photochemical reaction is the most obvious from noon to afternoon and in summer. In addition, the degree of air pollution generated in this area is also the most serious from noon to afternoon. As a result, $O_3$ and $PM_{10}$ concentration are usually higher during this time.

In regard to the analysis of $\gamma_{i,t}$ in the EGARCH model, the t-statistic of $\gamma_1$ in the VARMA (1,1)-EGARCH (1,1) model was a significant at 3.13, with a value of −0.082 (<0). This result suggests that when the concentrations of $PM_{10}$ and $O_3$ change drastically in winter, air quality in Kaohsiung and Pingtung will become noncompliant with the air quality standard. The Kaohsiung–Pingtung area is susceptible to air pollution because it is a populous industrial hub located in southern Taiwan. It has a large number of cars and scooters, as well as facilities that release pollutants regularly (namely chimneys). During winter, air quality is particularly prone to being noncompliant with the air quality standard set by the Environmental Protection Administration due to high $PM_{10}$ concentration. On the other hand, air pollutants quickly disperse in the air in summer; thus, there are fewer days with poor air quality than in the winter. This phenomenon reflects the leverage effect (when $\gamma_1 < 0$) in the EGARCH model [20]. Taiwan has done a magnificent job in containing COVID-19 and has received critical acclaim globally. The pandemic barely has any impact on businesses and industries in the nation. According to statistics up to the end of August 2020, there were only 493 confirmed cases, and only seven people died from COVID-19, a tiny fraction of the nation's 23 million population. The monitoring stations mentioned in this study were still operating during the pandemic and were not compromised by COVID-19 at all. That is, during this period, the monitoring results did not show any abnormal phenomena.

**Table 5.** Parameter estimation during the process when photochemical pollution factor VARMA (1,1) was paired with EGARCH (1,1).

| Vector Model | $a_0$ | $a_1$ | $a_2$ | $b_0$ | $b_1$ | $b_2$ | $c_0$ | $c_1$ | $c_2$ | $d_1$ | $\alpha_0$ | $\alpha_1$ | $\alpha_2$ | $\beta_1$ | $\gamma_1$ |
|---|---|---|---|---|---|---|---|---|---|---|---|---|---|---|---|
| VARMA (1,0) | 0.96 | −2.31 | | 3.143 | 1.03 | | 0.51 | 0.62 | −1.12 | 0.18 | 2.63 | −1.69 | 0.56 | 0.26 | −0.163 |
| t-statistic | 1.24 | 0.96 | | −0.77 | 2.14 | | −3.14 | −1.55 | 0.78 | 0.66 | 0.53 | 3.22 | 1.16 | 4.55 | 4.42 |
| VARMA (2,0) | −3.44 | 1.99 | | 0.87 | 3.46 | −2.59 | 0.97 | −1.52 | 0.53 | 1.32 | 0.64 | 2.21 | 1.58 | 0.33 | −0.035 |
| t-statistic | −1.25 | 2.55 | | 1.57 | −1.96 | 1.41 | 2.64 | 0.32 | 1.63 | 0.09 | −3.11 | 1.14 | −0.76 | 2.14 | 2.56 |
| VARMA (0,1) | 1.97 | | | 1.14 | 1.25 | 2.14 | 0.16 | 3.02 | 1.19 | 1.84 | 3.21 | −0.34 | 1.03 | 2.3 | −0.321 |
| t-statistic | 0.94 | | | −3.63 | −0.88 | 2.45 | −1.16 | 2.46 | 1.51 | −0.71 | 2.65 | 0.09 | −2.03 | 1.99 | 1.93 |
| VARMA (0,2) | 2.01 | | 0.96 | 3.03 | | | 1.13 | 0.06 | 1.59 | −0.16 | 0.92 | 0.56 | 0.31 | 1.55 | |
| t-statistic | 2.51 | | 0.94 | 2.17 | | | 0.88 | −2.12 | 3.12 | 1.22 | 1.17 | −2.77 | −0.26 | 4.02 | |
| VARMA (2,1) | −0.29 | 2.016 | 2.23 | 2.31 | 1.3 | 3.01 | −0.63 | 1.59 | 0.86 | 2.07 | −0.17 | 2.03 | 1.14 | −1.67 | |
| t-statistic | 1.38 | 3.46 | −1.65 | −1.10 | 2.05 | 1.16 | 1.54 | 2.62 | 1.14 | 2.63 | −0.33 | −1.68 | 0.82 | 2.31 | |
| VARMA (1,1) | 1.37 | 1.13 | 5.14 | −2.42 | 0.07 | 3.41 | 0.96 | 2.19 | 0.17 | 1.56 | 5.68 | 0.71 | 0.52 | 5.26 | −0.082 |
| t-statistic | 0.31 | 7.03 | 1.14 | −0.75 | 2.56 | 2.04 | 0.32 | 3.79 | 2.92 | 3.41 | 3.02 | 2.65 | 0.7 | 8.16 | 3.13 |

Note: (1) $O_3 = a_o + a_1 O_{3(t-1)} + a_2 O_{3(t-2)} + b_0 PM_{10(t)} + b_1 PM_{10(t-1)} + b_2 PM_{10(t-2)} + c_0 NO_{2(t)} + c_1 NO_{2(t-1)} + c_2 NO_{2(t-2)} + d_1 \varepsilon_{t-1}$ $h_t = \alpha_0 + \alpha_1 \varepsilon_{t-1}^2 + \alpha_2 \varepsilon_{t-2}^2 + \beta_1 h_{t-1}$ with $\varepsilon_t \sim N(0,1)$. A large $\varepsilon_{t-1}^2$ or $h_{t-1}$ gives rise to a large $\sigma_t^2$. This means that a large $\varepsilon_{t-1}^2$ tends to be followed by another large $\varepsilon_t^2$, generating, again, the well-known behavior of volatility clustering in time series. (2) $\gamma_{i,t} = c_{i0} + \sum_{j=1}^{p_i} b_{ij} \gamma_{i,t-j} + u_{i,t}$. (3) If $\gamma_i = 0$, it indicates that air quality has a symmetrical effect on the reaction to news shock. If $\gamma_i < 0$, it suggests that air quality has a leverage effect on the reaction to news shock. (4) A t-statistic value ≥ 1.96 means that the parameter reaches the significance level. In contrast, a t-statistic value < 1.96 suggests that the parameter does not reach the significance level.

### 3.4. The Predictive Power of the VARMA (1,1)-EGARCH (1,1) Model on the Three Air Pollutants

Based on an impact responses analysis, we applied the VARMA (1,1)-EGARCH (1,1) model to estimate the three air pollutants of the photochemical pollution factor, in order to test the predictive power and degree of the model under the influence of leverage effect (Table 5 shows that $\gamma_1 < 0$). Figures 2–4 display the estimated concentrations of the three air pollutants with the first 400 data from the sequence. The estimation results indicated that the correlation coefficients (r) of the predictive power of $PM_{10}$, $O_3$, and $NO_2$ were 0.832, 0.814, and 0.773, respectively, fully reflecting the analysis results mentioned previously (Kuo and Ho, 2018). In other words, the concentration of $NO_2$ did not change as drastically as the other two pollutants because $NO_2$ was less affected by the change of seasons, suggesting that it had the strongest asymmetric effect, or $NO_2$ has the least influence on the GARCH effect when its concentration fluctuations are triggered by the change of seasons, which makes it more difficult to predict its concentration fluctuations. Moreover, the EGARCH model could effectively reduce bias resulting from conditional heteroskedasticity when estimating correlation coefficients. The model enabled the presentation of diverse forms when it comes to dynamic variance modeling, and could accurately estimate conditional variances among the three air pollutants.

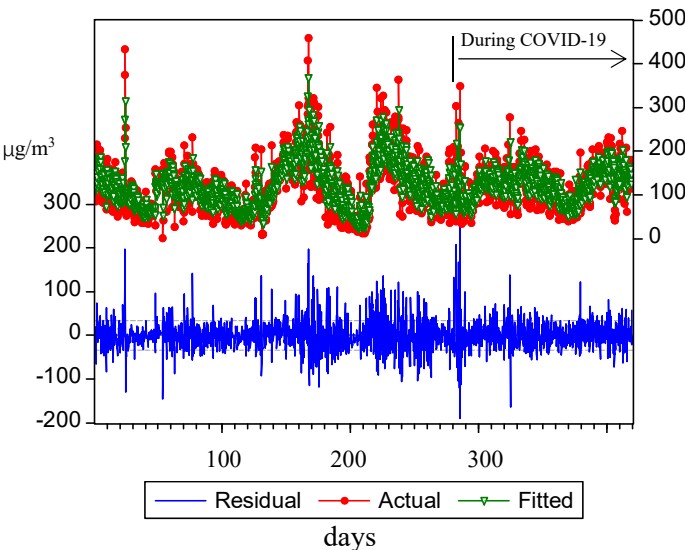

**Figure 2.** Estimation of the concentration of $PM_{10}$ with the VARMA (1,1)-EGARCH (1,1) model.

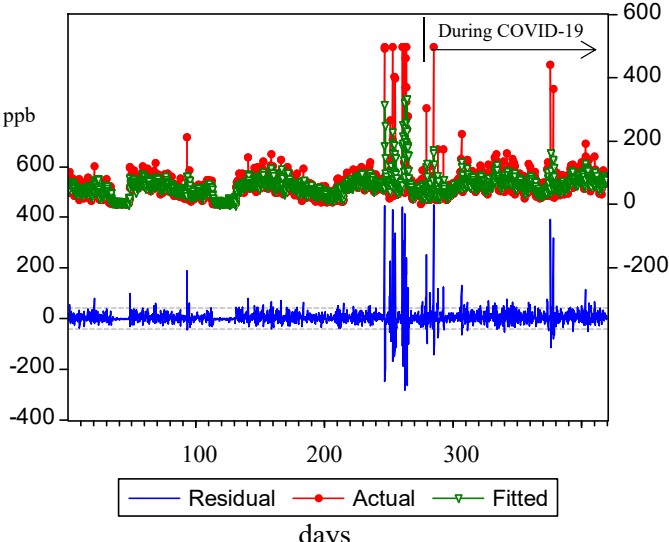

**Figure 3.** Estimation of the concentration of $O_3$ with the VARMA (1,1)-EGARCH (1,1) model.

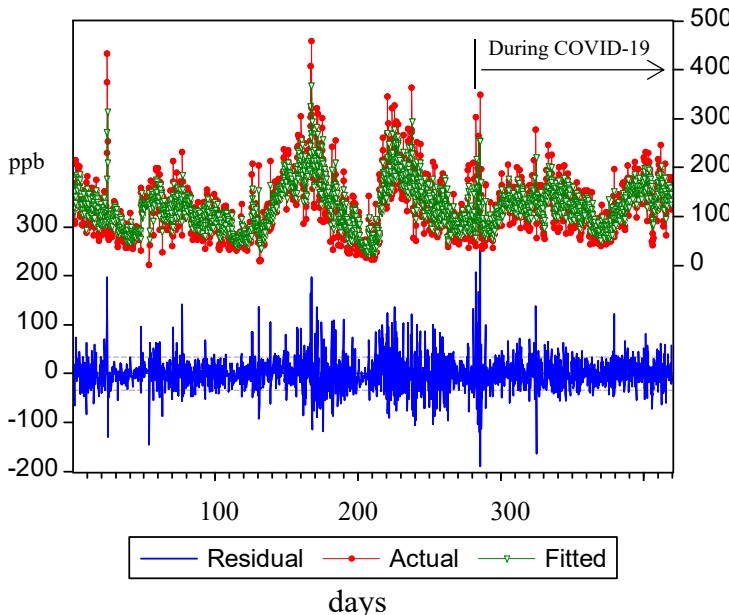

**Figure 4.** Estimation of the concentration of $NO_2$ with the VARMA (1,1)-EGARCH (1,1) model.

## 4. Conclusions

This study adopted the VARMA-EGARCH model to explore the changes and influences of variation of air pollutants in the time series. This approach is unprecedented because there are barely any published studies on how multiple air pollutants change and influence each other in an Air Pollutant Control Area with the EGARCH model. Moreover, since the parameters of the traditional multivariate GARCH model are too complex, while the multivariate GARCH model cannot fully represent multivariate analysis results, we applied dummy variables from the EGARCH model to the variant formula to enable it to accurately estimate the variances of the three air pollutants and capture the seasonal fluctuations of air quality in the VARMA-EGARCH model. This study also applied the VARMA (1,1)-EGARCH (1,1) model to explore the relativity and trend of concentration of $PM_{10}$, $O_3$, and $NO_2$ in the time series. When $PM_{10}$ and $NO_2$ form in the atmosphere, their concentrations cannot be used to estimate the concentration of $O_3$ in the current term. Rather, we must wait until they have been involved in photochemical reactions for a while, as that is when $O_3$ starts to form. In other words, only $PM_{10}$ and $NO_2$ with one- or two-term lag can influence the formation of $O_3$ in the current term; one-term lagged $PM_{10}$ and $NO_2$ have stronger impacts on the concentration of $O_3$ than those with a lag time of two terms. Finally, the estimation of the three air pollutants using the VARMA (1,1)-EGARCH (1,1) model indicated that the concentration of $NO_2$ did not change as drastically as the other two pollutants because $NO_2$ was less affected by the change of seasons. This suggested that since $NO_2$ had the strongest asymmetric effect, it was thus more difficult to predict its concentration fluctuations.

We adopted the Pollution Standards Index for evaluating the degree of air pollution. On the other hand, the Air Pollution Index (API) is based on the Ambient Air Quality Standard GB 3095-1996 and only assesses $SO_2$, $NO_2$, and $PM_{10}$. It was abolished and supplanted by the Air Quality Index (AQI), which is based on GB 3095-2012 and evaluates $SO_2$, $NO_2$, $PM_{10}$, $PM_{2.5}$, $O_3$, and CO; the AQI is mainly adopted for studying the degree of influence of air quality on human health. We did not study the variation of $PM_{2.5}$ concentration because $PM_{2.5}$ was included in $PM_{10}$; the subject of this study was not air pollutants' impact on human health, but rather the formation mechanism of various air pollutants and the interplay among these pollutants over time.

In this study, the EGARCH model made it possible to track the degree of change of air pollutants in each time series in real time. It also took the heterogeneity of each air pollutant variant into consideration, an aspect ignored in past studies. The research findings can serve as a reference for the government when it comes to the application and certification of air quality models, simulation

of models for a maximum allowable limit of increments, evaluation of air quality improvement schemes, and planning of strategies. In addition, the results of this study can also be referenced by authorities for planning air quality total quantity control, applying and examining various air quality models, simulating the allowable increase in air quality limits, and evaluating the benefit of air quality improvement. Furthermore, we will refer to the API and AQI methods if we are going to delve deeper into studies related to air pollution in the future and could be useful in the future extension of their work.

**Author Contributions:** E.M.-Y.W. formulates the research of direction of this study clearly. In addition, he collects and compiles data of ten air quality monitoring stations in the Kaohsiung–Pingtung Air Pollutant Control Area. S.-L.K. mainly carries out the execution of EVIEWS 10.0 and evaluates the variations of photochemical pollution factors and the pattern of mutual influences for the various air pollution species with respect to time series. All authors have read and agreed to the published version of the manuscript.

**Funding:** This research received no external funding.

**Conflicts of Interest:** The authors declare no conflict of interest.

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
