# Peer review of "VARMA-EGARCH Model for Air-Quality Analyses and Application in Southern Taiwan"

_atmosphere, doi:10.3390/atmos11101096_

Round 1

Reviewer 1 Report

The title contains twice the nouns “air” and “quality” as well as “quality”. The authors should use “air-quality” and simplify the title targeting their work. Total quality control is inappropriate. Perhaps the authors mean areas where regulatory interventions are required?

Ln.11. “Analyze” with what, numerical model, statistical analysis, measured campaign or empirical knowledge?

Ln.11. “What are pollution factors”?

Ln.15. Why this region? Is there anything special for selecting this part of the world?

Ln.30. The asymmetry effect is not clear. Please become specific or consider shortening this abstract that is rather long. The abstract should contain what are the key findings and why this work is “unique and merits publication”.

Ln.38-39. The use of hyphens is necessary, also EGARCH is not a keyword of the English dictionary but rather a tool used for a specific group of scientists (perhaps not necessary for the keywords).

Ln.49. Instead of diffusive use the term “dispersive or long range transported”.

Ln.57-60. The indices AQI and API are “indicators” that hide a lot of information instead of measurable pollutant concentrations. The authors should mention why they have used non-measurable indicators and perhaps should also consider the approach of “Air Quality Indicators for Uniform Indexing of Atmospheric Pollution over Large Metropolitan Areas. Atmospheric Environment, 1999, 33, 1861-1879”

Ln.73 to 79. This section is part of the methodology, where the domain should be properly presented. The reader expects at the introduction to read why this work is unique, what is new and worth to be published and why this part of the world was selected.

Ln.83 to 94. The google map is not a sufficient description. One needs to know the type of the station, the elevation, the climatological conditions and the instruments used for each pollutant and their calibration requirements. Of particular importance are the potential gaps in the recorded time series.

Ln.102 to 106. After 100 lines of text the reader does not know what are these factors. Are these something that is explained below?

Ln.206. Why these symbols are 1 line above the text. Are these superscripts?

Ln.234. Similar to Ln.234.

Ln.238. What is the significance of these statistical analysis and what is the expected number. Do the authors really need three decimal digits for all these numbers?

Ln.260 and 261. Why two decimal digits and what is the significance of the equations and symbols at the note.

Ln.330 to 334. Are these speculative or reality for the later why the authors have not demonstrated this with the historical data and the number of exceedances observed?

Ln.339-340. Similar to Ln.261 the equations should be fully explained.

Ln.363. Why of the to legends use the scale at the right hand and for which days of the year are observed the peaks between 160 and 280 in the x-scale. What are the values and their units at the axis of figure 2?

Ln.366. Similar to Ln.363.

Ln.369. Similar to Ln.363.

Ln.371 to 393. The reader expects to find why this work is essential and if it is valid in other years at the same site or in other sites. If those two aspects are not mentioned this work will be yet another statistical approach with limited generalisation values.

Therefore, I strongly suggest the drastic reformulation of this manuscript. I also strongly advise the authors to identify in their work the period in 2020 with the lockdown and social distancing restrictions for which the statistical mechanism should be different as globally observed from the satellite images.

Author Response

The title contains twice the nouns “air” and “quality” as well as “quality”. The authors should use “air-quality” and simplify the title targeting their work. Total quality control is inappropriate. Perhaps the authors mean areas where regulatory interventions are required?

Ans:(1)Thanks for your suggestion and they have been revised as instructed. The title has changed to “VARMA-EGARCH Model for Air-Quality Analyses and Application in Southern Taiwan”.

    (2)This study explored three air pollutants (PM10, O3 and NO2) belonging to the photochemical pollution factor, examining their relationships in the time series and variation trends of their concentrations. The air pollutant control area in the Kaohsiung-Pingtung area was chosen as its research subject because the region is generally severely affected by industrial pollution.

Ln.11. “Analyze” with what, numerical model, statistical analysis, measured campaign or empirical knowledge?

Ans: The EGARCH model was employed for statistical analysis; we adopted it to study the relationships among the concentrations of PM10, O3 and NO2 in the time series, as well as their variation trends. The EGARCH model has fewer limits than traditional GARCH models because its conditional variance after logarithmic transformation is always positive, regardless of whether or not the coefficient of the variance formula for estimation is positive.

Ln.11. “What are pollution factors”?

Ans: Prior to the EGARCH analysis, we selected seven kinds of air pollutants from the ten air quality monitoring stations in the Kaohsiung-Pingtung Air Pollutant Control Area through a multivariate statistical analysis, and then divided them into three kinds of factors via factor analysis: the organic pollution factor, photochemical pollution factor, and fuel factor. These three factors can represent the main factors affecting the air quality in the Kaohsiung-Pingtung Air Pollutant Control Area.

Ln.15. Why this region? Is there anything special for selecting this part of the world?

Ans: Kaohsiung City is the hub of Taiwan’s petrochemical industry and heavy industry. It suffers from the most severe air pollution in Taiwan due to tightly-packed industrial areas, a densely-located population, and the concentration of stationary and mobile pollution sources. The Environmental Protection Administration of Taiwan plans the nation’s air pollutant control areas based on terrains, as well as types of industry and climates, drawing municipalities, cities or counties between which air pollutants might circulate into an air pollutant control area; this is why Kaohsiung City and Pingtung County are in the same air pollutant control area.

Ln.30. The asymmetry effect is not clear. Please become specific or consider shortening this abstract that is rather long. The abstract should contain what are the key findings and why this work is “unique and merits publication”.

Ans:(1)The explanation of the asymmetry effect has been included in the abstract, which has also been simplified and updated with the purpose and contribution of this study.

(2)This study is the first of its kind in the world to explore the interplay among various air pollutants and reactions triggered by such interplay over time. We believe it represents a breakthrough in exploring the variation trends of air pollution in environmental engineering studies.

(3)The abstract became longer after a slight revision requested from a reviewer, who required us to elaborate on the reasons for conducting a multivariate statistical analysis.

Ln.38-39. The use of hyphens is necessary, also EGARCH is not a keyword of the English dictionary but rather a tool used for a specific group of scientists (perhaps not necessary for the keywords).

Ans: Thanks for your suggestion and the “EGARCH” of the keywords has been deleted.

Ln.49. Instead of diffusive use the term “dispersive or long range transported”.

Ans: Thanks for your suggestion and they have been revised as instructed.

Ln.57-60. The indices AQI and API are “indicators” that hide a lot of information instead of measurable pollutant concentrations. The authors should mention why they have used non-measurable indicators and perhaps should also consider the approach of “Air Quality Indicators for Uniform Indexing of Atmospheric Pollution over Large Metropolitan Areas. Atmospheric Environment, 1999, 33, 1861-1879”

Ans: Thanks for your suggestion.

  • We have explained why we adopted the Pollution Standards Index for evaluating the degree of air pollution. Please refer to the Introduction.
  • Taiwan’s air quality is quantified using a pollution standards index (PSI) with the results released to the public, following the procedure identical to that of the United States Environmental Protection Agency. The nation’s PSI was formulated and promulgated in 1994 by the Environmental Protection Administration based on the daily concentrations of five kinds of air pollutants: particulate matter with particles below 10 microns (PM10), sulfur dioxide (SO2), nitrogen dioxide (NO2), carbon monoxide (CO), and ozone (O3).
  • The Air Pollution Index (API) is based on the Ambient Air Quality Standard GB 3095-1996 and only assesses SO2, NO2, and PM10. It was abolished and supplanted by the Air Quality Index (AQI), which is based on GB 3095-2012 and evaluates SO2, NO2, PM10, PM5, O3, and CO; the AQI is mainly adopted for studying the degree of influence of air quality on human health. We did not study the variation of PM2.5 concentration because PM2.5 was included in PM10; the subject of this study was not air pollutants’ impact on human health, but rather the formation mechanism of various air pollutants and the interplay among these pollutants over time.
  • We will refer to the reviewer’s references if we are going to dive deeper into studies related to air pollution in the future.

Ln.73 to 79. This section is part of the methodology, where the domain should be properly presented. The reader expects at the introduction to read why this work is unique, what is new and worth to be published and why this part of the world was selected.

Ans: Thank you for your suggestion.

  • Following your previous suggestion (Ln.30. The asymmetry effect…), we have already added a note to the abstract that this study is the first of its kind in the world to explore the interplay among various air pollutants and reactions triggered by such interplay over time. We believe it represents a breakthrough in exploring the variation trends of air pollution in environmental engineering studies.
  • We have already included in the last paragraph of the Introduction suggestions for applying this study’s findings in future air quality studies and this study’s contributions.

Ln.83 to 94. The google map is not a sufficient description. One needs to know the type of the station, the elevation, the climatological conditions and the instruments used for each pollutant and their calibration requirements. Of particular importance are the potential gaps in the recorded time series.

Ans: Revisions of section 2.1 have been made according to the reviewer’s suggestion.

Ln.102 to 106. After 100 lines of text the reader does not know what are these factors. Are these something that is explained below?

Ans: A description of grouping pollutants into the photochemical pollution factor, fuel pollution factor, and organic pollution factor via a multivariate statistical analysis has been added to section 2.2. Among the three factors, the photochemical pollution factor was chosen as the subject of this study.

Ln.206. Why these symbols are 1 line above the text. Are these superscripts?

Ans: These symbols are not superscripts. They have already been revised.

Ln.234. Similar to Ln.206.

Ans: Thanks for your suggestion and authors have completed the correction.

Ln.238. What is the significance of these statistical analysis and what is the expected number. Do the authors really need three decimal digits for all these numbers?

Ans:(1) The items in Table 1 (such as kurtosis, skewness...) mainly discuss the characteristics of the O3, PM10, and NO2 series after a long period of time. For example, through statistical analysis, high kurtosis means that the increase in variance is mainly caused by extreme differences in low frequencies that are greater than or less than the average. Skewness is mainly a measure of the direction and degree of skewness of statistical data distribution. Through the analysis of these statistical tools, we can further understand the statistical significance of the sequence of numbers.

   (2) Thanks for your valuable suggestion。Statistics in Table 1 have all been revised to be rounded to the nearest integer and tenth.

Ln.260 and 261. Why two decimal digits and what is the significance of the equations and symbols at the note.

Ans:(1) We adopted EVIEWS 10.0 for the LB-Q test. The software produced critical values down to two decimal digits, so the statistics in this study also include those down to two decimal digits.

(2) The said note has been deleted. We have also included explanations about the LB-Q test to make the content more complete.

Ln.330 to 334. Are these speculative or reality for the later why the authors have not demonstrated this with the historical data and the number of exceedances observed?

Ans:(1) The description of the air quality of the Area in this study was real rather than speculative. The subject of our study, the Kaohsiung-Pingtung Air Pollutant Control Area, includes Kaohsiung City, which suffers from the most severe air pollution in Taiwan. The City, home to approximately 2.77 million people, has eight industrial areas and a large steel plant. Air quality is usually poor due to the industrial activities in those areas and the large number of cars and motorcycles in the region.

(2)The Kaohsiung-Pingtung Area is susceptible to pollutants from external sources brought by the north-eastern seasonal wind, and pollutants traveling from northern and central Taiwan. To make matters worse, pollutants are trapped in the Area by an inversion caused by the seasonal wind as it descends from the Central Mountain Range. In addition, emissions from factories and cars concentrated in Greater Kaohsiung make poor air quality more likely in the region during winter.

(3)  Base on the above reasons, the air pollution in this area has been quite serious over the years. The authors have added related descriptions in section 3.2.1.

Ln.339-340. Similar to Ln.261 the equations should be fully explained.

Ans: Thank you for your valuable comments.

  • Explanation of γi,t has been provided in Section 2.4.
  • An extra explanation is presented in original Ln.330. Please refer to Table 5 for more details.

Ln.363. Why of the to legends use the scale at the right hand and for which days of the year are observed the peaks between 160 and 280 in the x-scale. What are the values and their units at the axis of figure 2?

Ans:(1) Figure 2 was the result of an analysis conducted using EVIEWS 10.0. The peaks in the graph indicate higher concentrations of PM10 resulting from severe air pollution.

    (2) The vertical axis in Figure 2 represents units of the pollutants: μg/m3 for PM10, and ppb for O3 and NO2.

Ln.366. Similar to Ln.363.

Ans: The measurement unit for O3 is ppb.

Ln.369. Similar to Ln.363.

Ans: The measurement unit for NO2 is ppb.

Ln.371 to 393. The reader expects to find why this work is essential and if it is valid in other years at the same site or in other sites. If those two aspects are not mentioned this work will be yet another statistical approach with limited generalisation values.

Therefore, I strongly suggest the drastic reformulation of this manuscript. I also strongly advise the authors to identify in their work the period in 2020 with the lockdown and social distancing restrictions for which the statistical mechanism should be different as globally observed from the satellite images.

Ans: Thank you for your valuable comments.

  • Taiwan has done a magnificent job in containing COVID-19 and has received critical acclaim globally. The pandemic barely has any impact on businesses and industries in the nation. According to statistics up to the end of August 2020, there were only 493 confirmed cases, and only seven people died from COVID-19, a tiny fraction of the nation’s 23 million population. The monitoring stations mentioned in this study were still operating during the pandemic and were not compromised by COVID-19 at all.
  • We only had limited time for our research. However, since the Atmosphere invited us to contribute to their special issue, we spared no effort to collect data, plan the paper, and implement the chosen models. Also, because this research is the first to study the interplay among various air pollutants and reactions triggered by them over time via a VARMA-EGARCH model, it was inevitable that the application and study of the model went amiss in some ways. We will try to improve those aspects in the future.

Reviewer 2 Report

The manuscript corresponds to the direction of the special issue and is recommended for publication. Wish: to continue research using the statistical method proposed in the article, which takes into account physical and chemical processes.

Author Response

The manuscript corresponds to the direction of the special issue and is recommended for publication. Wish: to continue research using the statistical method proposed in the article, which takes into account physical and chemical processes.

Ans: Thank you very much for your affirmation. We will continue to conduct research on statistics analyses.

Reviewer 3 Report

Review Report

---Abstract should be more precise, so need to shorten it focusing on significance of the work.

---The sentence in line 21: "The research findings indicate that....." is confusing and need to be re-written.

---The first sentence of introduction section (Line 42) is not valid. The air quality monitoring report the authors were talking about was published in 1994. So it’s not a new report. But the authors have mentioned that it is a recent report. So this sentence should be changed.

---What is the reason for selecting PM10, O3, and NO2 for this work? Why the authors didn't consider PM2.5, which is more harmful to mankind and is a major pollutant that causes AQI value to increase in Taiwan?

---The statement in Line 58 needs to be corrected. AQI is not the other name for PSI as written by authors, both are different. So authors should read relevant literature and correct the statement accordingly.

---There is relatively high similarity for this manuscript with two of the authors previously published works (around 15% combined). This similarity should be avoided. So it is better if the sections 2.5.1, 2.5.3, 2.5.4, and paragraph 3 (Line 61-72) of the introduction section can be rewritten.

---Figure 1 is a google map image. So whether the authors have the right to use google maps in publications without due credit to Google? The authors need to confirm this. For your information, it is possible to use softwares like QGIS to create maps and use in journal publications.

Is there any reason for selecting Kaohsiung-Pingtung region for this study? If yes, please provide the details in section 2.1 of the manuscript.

It is understood that authors used factor analysis for multivariate statistics to analyze the three factors including photochemical pollution factor, the fuel pollution factor, and the organic pollution factor. How the authors know that the three factors that most influence air quality in the Kaohsiung-Pingtung area are photochemical pollution factor, the fuel pollution factor, and the organic pollution factor? Is it based on any study?

---How the authors reached that conclusion that NO2 was less effected by the change of seasons? Did the authors carried out seasonal variation analysis or is it based on any previous study results? Please explain.

----Most of the references cited by the authors are published 10 years back, especially regarding the air pollution. There will be changes in air pollution of a region based on various factors related to environment, Government policies etc. So it is always better to follow latest literature related to air pollution of the region of study and include it in the manuscript.  For instance, the statement by authors in the manuscript that PM10 and O3 were the major air pollutants causing PSI to exceed standards is based on an old report. The latest statistics related to air pollution in Taiwan can be included in introduction section. Authors can refer the recent research articles that deals with air pollution in Taiwan.

Hsu, Chia-Hua, and Fang-Yi Cheng. "Synoptic weather patterns and associated air pollution in Taiwan." Aerosol and Air Quality Research 19.5 (2019): 1139-1151.

Balram, Deepak, Kuang-Yow Lian, and Neethu Sebastian. "Air quality warning system based on a localized PM2. 5 soft sensor using a novel approach of Bayesian regularized neural network via forward feature selection." Ecotoxicology and environmental safety 182 (2019): 109386.

Maurer, Maylin, et al. "Trends of Fog and Visibility in Taiwan: Climate Change or Air Quality Improvement?." Aerosol and Air Quality Research 19.4 (2019): 896-910+.

Balram, Deepak, Kuang-Yow Lian, and Neethu Sebastian. "A novel soft sensor based warning system for hazardous ground-level ozone using advanced damped least squares neural network." Ecotoxicology and Environmental Safety 205 (2020): 111168.

Author Response

---Abstract should be more precise, so need to shorten it focusing on significance of the work.

Ans: Thanks for your suggestion and they have been revised as instructed.

---The sentence in line 21: "The research findings indicate that....." is confusing and need to be re-written.

Ans: This part has been revised according to the reviewer’s suggestion. Please refer to the abstract for details.

---The first sentence of introduction section (Line 42) is not valid. The air quality monitoring report the authors were talking about was published in 1994. So it’s not a new report. But the authors have mentioned that it is a recent report. So this sentence should be changed.

Ans: Thank you for pointing out the mistake. The word “recent” has been removed to prevent confusion.

---What is the reason for selecting PM10, O3, and NO2 for this work? Why the authors didn't consider PM2.5, which is more harmful to mankind and is a major pollutant that causes AQI value to increase in Taiwan?

Ans: In Taiwan, air quality data are released to the public in the form of the Pollution Standards Index (PSI) values, following the same system adopted by the United States Environmental Protection Agency. Since the PSI includes PM10, SO2, NO2, CO, O3, but not PM2.5, we decided to exclude the variation of PM2.5 concentration from this study. Also, because PM2.5 is included in PM10, and this study’s subjects are the formation mechanism of various air pollutants and their interplay over time, rather than air pollutants’ threat to health, we did not see any reason to include PM2.5 in this study.

---The statement in Line 58 needs to be corrected. AQI is not the other name for PSI as written by authors, both are different. So authors should read relevant literature and correct the statement accordingly.

Ans: The authors have corrected the content and the statement of AQI has been delected.

---There is relatively high similarity for this manuscript with two of the authors previously published works (around 15% combined). This similarity should be avoided. So it is better if the sections 2.5.1, 2.5.3, 2.5.4, and paragraph 3 (Line 61-72) of the introduction section can be rewritten.

Ans: Sections 2.5.1, 2.5.3, 2.5.4, and paragraph 3 in the Introduction have all been revised.

---Figure 1 is a google map image. So whether the authors have the right to use google maps in publications without due credit to Google? The authors need to confirm this. For your information, it is possible to use softwares like QGIS to create maps and use in journal publications.

Ans: We believe that using Google Maps images without giving credit to Google will not be a problem. We have checked many SCI journals and found many using images from the service. We have also confirmed that there is no copyright issue for using Google Maps images. In terms of QGIS, we did not use it this time because we had neither the knowledge nor experience in using the said software. However, we will start learning it now so that we can use it in our next research.

Is there any reason for selecting Kaohsiung-Pingtung region for this study? If yes, please provide the details in section 2.1 of the manuscript.

Ans: Thanks for your suggestion and they have been provided as instructed.

It is understood that authors used factor analysis for multivariate statistics to analyze the three factors including photochemical pollution factor, the fuel pollution factor, and the organic pollution factor. How the authors know that the three factors that most influence air quality in the Kaohsiung-Pingtung area are photochemical pollution factor, the fuel pollution factor, and the organic pollution factor? Is it based on any study?

Ans: We divided the seven air pollutants into three groups via a multivariate statistical analysis: the photochemical pollution factor, fuel pollution factor, and organic pollution factor. The initial eigenvalues of the three factors were 1.982, 1.526, and 1.201, respectively. The higher the eigenvalue, the higher the percentage of air affected by a factor and the stronger the impact of a factor on air quality. In other words, the factor with the highest initial eigenvalue could best present the degree of variation of air quality in the area.

---How the authors reached that conclusion that NO2 was less effected by the change of seasons? Did the authors carried out seasonal variation analysis or is it based on any previous study results? Please explain.

Ans: According to the relevant statistics, the major factor causing the air quality in the area to exceed the PSI was PM10, followed by O3; the NO2 concentration always remained below the standard of causing air pollution. Furthermore, from the conversion table of PSI and pollutant concentration, one can see that there is no corresponding PSI value for NO2 concentration between 0~599 ppb; only after the NO2 concentration reaches 600 ppb can one find a corresponding PSI value (100). Therefore, we can say that NO2 was least affected by the change of seasons in the photochemical pollution factor.

----Most of the references cited by the authors are published 10 years back, especially regarding the air pollution. There will be changes in air pollution of a region based on various factors related to environment, Government policies etc. So it is always better to follow latest literature related to air pollution of the region of study and include it in the manuscript.  For instance, the statement by authors in the manuscript that PM10 and O3 were the major air pollutants causing PSI to exceed standards is based on an old report. The latest statistics related to air pollution in Taiwan can be included in introduction section. Authors can refer the recent research articles that deals with air pollution in Taiwan.

Hsu, Chia-Hua, and Fang-Yi Cheng. "Synoptic weather patterns and associated air pollution in Taiwan." Aerosol and Air Quality Research 19.5 (2019): 1139-1151.

Balram, Deepak, Kuang-Yow Lian, and Neethu Sebastian. "Air quality warning system based on a localized PM2. 5 soft sensor using a novel approach of Bayesian regularized neural network via forward feature selection." Ecotoxicology and environmental safety 182 (2019): 109386.

Maurer, Maylin, et al. "Trends of Fog and Visibility in Taiwan: Climate Change or Air Quality Improvement?." Aerosol and Air Quality Research 19.4 (2019): 896-910+.

Balram, Deepak, Kuang-Yow Lian, and Neethu Sebastian. "A novel soft sensor based warning system for hazardous ground-level ozone using advanced damped least squares neural network." Ecotoxicology and Environmental Safety 205 (2020): 111168.

 Ans: We appreciate the suggestion. The above references have been cited in this study.

Reviewer 4 Report

The paper describes a set of statistical analyses aimed at applying what the authors claim a new statistical approach to environmental time series.

The authors propose a large number of equations, graphs and a set of statistical correlations but only focus upon statistics and even the conclusions seem a mere list of statistical findings. Today we have a number of models describing the chemistry and physics of primary and secondary pollutants including the interactions between the three analyzed chemicals. The authors do not mention them at all and seem only to focus on their own study. Also, there are a plethora of papers describing the results of similar statistical studies on environmental time series, including ARMA, ARMAX, NN, and many others, but the authors prefer to refer to econometrics and mention very little to make a true comparative test.

The paper shows several weaknesses. Such weaknesses are testified by the fact that the authors do not compare their findings with other studies and do not support their methodology and conclusions with proper references. I think that the paper should be rewritten supporting findings, results and conclusions with proper literature. Some main other points follow.

Between, the authors keep mentioning a "photochemical pollution factor" of whom I never heard. It is very unusual but, anyhow, they must better describe it and the various terms, indices, and variables even if this requires twice the pages.

Another crucial problem arise from the fact that the authors seem to apply their models over daily average of pollution level (I must admit that I am not sure but fig.2-4 seem to indicate this). It is well known that, at least for photochemical smog, the daily average has very little significance and do not let understanding, for example, the Sillman indicator that, in turn, is crucial for understanding the true evolution of phenomena. A reference to this is given within conclusions.

Another concern I have regards the results of analysis. Fig.2-4 (which should be much larger) show rather poor results, being several deviations of the same order of time series.

Finally I have a question to pose. The authors claim that the time series are related to Jan19->May20 but the time series are not labelled so we cannot understand their timing. However, nothing shows the well-known effects of Covid pandemic that caused an abrupt sharp decrease of pollution levels worldwide. I know that Taiwan was one of the few countries to avoid a complete lockdown but I would have expected an unusual behavior of time series during pandemic, anyhow. Thus I am asking the authors at least to better label data and, if possible, to explain this.

Given the amount of work done and the sake of statistical correlations, I think that the paper could be accepted after another resubmission better explaining statistical findings also introducing a proper literature comparison. The models and the theory should be explained better also from environmental point of view. Please also enhance figure resolution and size.

Author Response

The paper describes a set of statistical analyses aimed at applying what the authors claim a new statistical approach to environmental time series.

--The authors propose a large number of equations, graphs and a set of statistical correlations but only focus upon statistics and even the conclusions seem a mere list of statistical findings. Today we have a number of models describing the chemistry and physics of primary and secondary pollutants including the interactions between the three analyzed chemicals. The authors do not mention them at all and seem only to focus on their own study. Also, there are a plethora of papers describing the results of similar statistical studies on environmental time series, including ARMA, ARMAX, NN, and many others, but the authors prefer to refer to econometrics and mention very little to make a true comparative test.

Ans: Thanks for your valuable suggestion. At present, EGARCH mode is almost always used in measurement Economic discussion. This research is the first of its kind in the world to adopt a VARMA-EGARCH model to explore the interplay among various air pollutants and reactions triggered by it over time. The results of this study can be referenced by authorities for planning air quality total quantity control, applying and examining various air quality models, simulating the allowable increase of air quality limits, and evaluating the benefit of air quality improvement.

--The paper shows several weaknesses. Such weaknesses are testified by the fact that the authors do not compare their findings with other studies and do not support their methodology and conclusions with proper references. I think that the paper should be rewritten supporting findings, results and conclusions with proper literature. Some main other points follow.

Ans: Thanks for your valuable suggestion. The authors have strengthened the article's discussion and content, and rewrite in each chapter. For example, We add the content “The results of this study can also be referenced by authorities for planning air quality total quantity control, applying and examining various air quality models, simulating the allowable increase of air quality limits, and evaluating the benefit of air quality improvement.” in Conclusion.

--Between, the authors keep mentioning a "photochemical pollution factor" of whom I never heard. It is very unusual but, anyhow, they must better describe it and the various terms, indices, and variables even if this requires twice the pages.

Ans: Thanks for your valuable suggestion. The author has made corrections in the content. We divided the seven air pollutants into three groups via a multivariate statistical analysis: the photochemical pollution factor, fuel pollution factor, and organic pollution factor. The initial eigenvalues of the three factors were 1.982, 1.526, and 1.201, respectively. The higher the eigenvalue, the higher the percentage of air affected by a factor and the stronger the impact of a factor on air quality. In other words, the factor with the highest initial eigenvalue could best present the degree of variation of air quality in the area.

--Another crucial problem arise from the fact that the authors seem to apply their models over daily average of pollution level (I must admit that I am not sure but fig.2-4 seem to indicate this). It is well known that, at least for photochemical smog, the daily average has very little significance and do not let understanding, for example, the Sillman indicator that, in turn, is crucial for understanding the true evolution of phenomena. A reference to this is given within conclusions.

Another concern I have regards the results of analysis. Fig.2-4 (which should be much larger) show rather poor results, being several deviations of the same order of time series.

Ans: We only had limited time for our research. However, since the Atmosphere invited us to contribute to their special issue, we spared no effort to collect data, plan the paper, and implement the chosen models. Also, because this research is the first to study the interplay among various air pollutants and reactions triggered by them over time via a VARMA-EGARCH model, it was inevitable that the application and study of the model went amiss in some ways. We will try to improve those aspects in the future.

Finally I have a question to pose. The authors claim that the time series are related to Jan19->May20 but the time series are not labelled so we cannot understand their timing. However, nothing shows the well-known effects of Covid pandemic that caused an abrupt sharp decrease of pollution levels worldwide. I know that Taiwan was one of the few countries to avoid a complete lockdown but I would have expected an unusual behavior of time series during pandemic, anyhow. Thus I am asking the authors at least to better label data and, if possible, to explain this.

Ans: Thanks for your valuable suggestion.

(1) Taiwan has done a magnificent job in containing COVID-19 and has received critical acclaim globally. The pandemic barely has any impact on businesses and industries in the nation. According to statistics up to the end of August 2020, there were only 493 confirmed cases, and only seven people died from COVID-19, a tiny fraction of the nation’s 23 million population. The monitoring stations mentioned in this study were still operating during the pandemic and were not compromised by COVID-19 at all.

  • The author has marked the time of occurrence of COVID-19 in Figure2~4. During this period, the monitoring results did not show any abnormal phenomena.

Given the amount of work done and the sake of statistical correlations, I think that the paper could be accepted after another resubmission better explaining statistical findings also introducing a proper literature comparison. The models and the theory should be explained better also from environmental point of view. Please also enhance figure resolution and size.

Ans: Thank you very much for your affirmation.

Round 2

Reviewer 1 Report

The authors did an effort to comply with the comments from reviewers 3 and 4. To some aspects, there have been improvements.

Unfortunately, in their approach they think that the review process is a dialogue with the reviewers and the answers provided by the authors are not accompanied with appropriate changes in the text. Hence, the readers of the journal must consult the main text plus the answers given to the two reviewers in order to have a better understanding. This approach is absurd especially since the text reviews is not automatically going to be published as open source text.

It is inexplicable why the units are still missing from the tables and figures and why one to three decimal digits are required at the tables.

Also, when the reviewers are providing a better reference to the text, it is expected that the authors acknowledge in the text that there are better indicator approaches (even mentioning them in the references) and then saying that these could be useful in future extension of their work.

For these reasons, I am "rejecting this work in its present" form. Surely, I expect that all the answers to reviewers are incorporated in the main text of this manuscript; the tables and figures become scientifically sound (appropriate units, detailed captions and right number of significant digits); explanations if the monitoring sites are urban, sub-urban or industrial).

Also, in the description of the operating domain must be inserted detailed climatological data that could help the reader to understand if the photochemical pollution is due to natural background or due to human emissions. I am sorry but these are the baring minimum considerations after reading the version 2 of this proposed publication.     

Author Response

--The authors did an effort to comply with the comments from reviewers 3 and 4. To some aspects, there have been improvements.

Ans: Thanks for your approval.

--Unfortunately, in their approach they think that the review process is a dialogue with the reviewers and the answers provided by the authors are not accompanied with appropriate changes in the text. Hence, the readers of the journal must consult the main text plus the answers given to the two reviewers in order to have a better understanding. This approach is absurd especially since the text reviews is not automatically going to be published as open source text.

Ans: Thanks for your valuable suggestion.

  • Before conducting this study, we selected seven kinds of air pollutants from the ten air quality monitoring stations in the Kaohsiung-Pingtung Air Pollutant Control Area through a multivariate statistical analysis, and then divided them into three kinds of factors via factor analysis: photochemical pollution factors, organic pollution factors, and fuel factors. These three factors can represent the main factors affecting the air quality in the Kaohsiung-Pingtung Air Pollutant Control Area. Factor with the most significant is then selected as the targets for conducting investigations in this study; it is termed “photochemical pollution factors”. We hope if this research is accepted, we will continue to discuss the application about multivariate statistical methods.
  • Currently, there are very few papers, if any, covers the application of the (E)ARCH mode in environmental engineering and related topics, especially on air pollution estimation and analyses. The analysis results can reflect the relationships among air pollutants in real time, the authorities can refer to the results when applying and examining various air quality models, simulating the allowable increase of air quality limits, and evaluating the benefits of air quality improvement.

--It is inexplicable why the units are still missing from the tables and figures and why one to three decimal digits are required at the tables.

Ans: Thanks for your suggestion.

  • Authors have added the units in Tables 1~3 and Figures 2~4.
  • Authors take the effective digits according to the execution result of EVIEWS 10.0 and the references. The first decimal place is almost used in Tables 1ï¼›nevertheless, the effective digits of Jarque-Bera are integer. In PM10, the effective digits of concentration are also integer.
  • In Table 2~3, many references take the effective digits of critical value to second decimal place.

--Also, when the reviewers are providing a better reference to the text, it is expected that the authors acknowledge in the text that there are better indicator approaches (even mentioning them in the references) and then saying that these could be useful in future extension of their work.

Ans: Thanks for your valuable suggestion and the conclusion has been revised to accommodate your suggestions.

--For these reasons, I am "rejecting this work in its present" form. Surely, I expect that all the answers to reviewers are incorporated in the main text of this manuscript; the tables and figures become scientifically sound (appropriate units, detailed captions and right number of significant digits); explanations if the monitoring sites are urban, sub-urban or industrial).

Ans: Thanks for your valuable suggestion and they have been revised as instructed.

(1)In this study, in these ten ordinary air quality monitoring stations, Meinong, Pingtung, Daliao, and Chaochou are located in sub-urban belonging to non-industrial zone; Nanzi, Renwu, Zuoying, Xiaogang and Linyuan belong to industrial zone; Cianjin is located in urban. We have added the contents in 2.1.

--Also, in the description of the operating domain must be inserted detailed climatological data that could help the reader to understand if the photochemical pollution is due to natural background or due to human emissions. I am sorry but these are the baring minimum considerations after reading the version 2 of this proposed publication.     

Ans: Thanks for your valuable suggestion and they have been revised as instructed in 3.3. Kaohsiung-Pingtung Air Pollutant Control Area is located in the tropics. The annual average temperature is larger than 20℃, and the photochemical reaction is most obvious from noon to afternoon and in summer. In addition, the degree of air pollution generated in this area is also the most serious from noon to afternoon. As a result, O3 and PM10 concentration are usually higher during this time.
